# Bacteriophage SRD2021 Recognizing Capsular Polysaccharide Shows Therapeutic Potential in Serotype K47 *Klebsiella pneumoniae* Infections

**DOI:** 10.3390/antibiotics10080894

**Published:** 2021-07-22

**Authors:** Guijuan Hao, Rundong Shu, Liqin Ding, Xia Chen, Yonghao Miao, Jiaqi Wu, Haijian Zhou, Hui Wang

**Affiliations:** 1Department of Microbiology, Nanjing Agricultural University, Nanjing 210095, China; 2015216024@njau.edu.cn (G.H.); 2018116064@njau.edu.cn (R.S.); 2019116064@njau.edu.cn (L.D.); xiachen@pku.edu.cn (X.C.); 2019116076@njau.edu.cn (Y.M.); 23217212@njau.edu.cn (J.W.); 2Department of Preventive Veterinary Medicine, College of Animal Science and Technology, Shandong Agricultural University, Taian 271018, China; 3Peking-Tsinghua Joint Center for Life Sciences, Academy for Advanced Interdisciplinary Studies, Peking University, Beijing 100871, China; 4State Key Laboratory for Infectious Disease Prevention and Control, Chinese Center for Disease Control and Prevention, Beijing 102206, China; zhouhaijian@icdc.cn

**Keywords:** *Klebsiella pneumoniae*, capsular polysaccharide, K47 serotype, phage receptor, phage therapy

## Abstract

*Klebsiella pneumoniae* is an opportunistic pathogen posing an urgent threat to global public health, and the capsule is necessary for *K. pneumoniae* infection and virulence. Phage-derived capsule depolymerases have shown great potential as antivirulence agents in treating carbapenem-resistant *K. pneumoniae* (CRKP) infections. However, the therapeutic potential of phages encoding depolymerases against CRKP remains poorly understood. In this study, we identified a long-tailed phage SRD2021 specific for mucoid CRKP with capsular K47 serotype, which is the predominant infectious K-type in Asia. Genome sequencing revealed that ΦSRD2021 belonged to the *Drulisvirus* genus and exhibited a capsular depolymerase domain in its tail fiber protein. A transposon-insertion library of host bacteria was constructed to identify the receptor for ΦSRD2021. We found that most phage-resistant mutants converted to a nonmucoid phenotype, including the mutant in *wza* gene essential for capsular polysaccharides export. Further knockout and complementation experiments confirmed that the Δ*wza* mutant avoided adsorption by ΦSRD2021, indicating that the K47 capsular polysaccharide is the necessary receptor for phage infection. ΦSRD2021 lysed the bacteria mature biofilms and showed a therapeutic effect on the prevention and treatment of CRKP infection in the *Galleria mellonella* model. Furthermore, ΦSRD2021 also reduced the colonized CRKP in mouse intestines significantly. By recognizing the host capsule as a receptor, our results showed that ΦSRD2021 may be used as a potential antibacterial agent for K47 serotype *K. pneumoniae* infections.

## 1. Introduction

*Klebsiella pneumoniae*, a Gram-negative bacterium of the *Enterobacteriaceae* family, is a common agent of global community-acquired and healthcare-associated infections. In humans, *K. pneumoniae* often colonize the gastrointestinal tract, skin, and nasopharynx. Two major types of antibiotic resistance, the expression of extended-spectrum β-lactamases (ESBLs) and carbapenemases, have been commonly observed in *K. pneumoniae* [1]. Due to the lack of available effective treatments, ESBL-producing and carbapenem-resistant *K. pneumoniae* infections have significantly higher rates of morbidity and mortality than infections with nonresistant bacteria. *K. pneumoniae* capsule is a well-studied virulence factor, consisting of an elaborate layer of surface-associated polysaccharides, termed K antigens. Genes that code for the biosynthesis of *K. pneumoniae* capsule are located on a chromosomal operon, *cps*, harboring a series of genes such as *wza*, *cpsB*, and *galF* [2,3]. It has been demonstrated that capsule polysaccharides (CPS) are critical for the resistance of *K. pneumoniae* to host defense, suppression of early inflammatory response, adherence, and biofilm formation [1]. Clinical statistics suggest that CPS produced by the infecting *K. pneumoniae* strain may be associated with the severity of infection, especially in community-acquired infections [1,4]. There are 78 K serotypes of *K. pneumoniae*, each of which produces a unique capsule. K1 and K2 serotype strains are generally more virulent than other serotypes; K47 and K64 are the most prevalent types in CRKP infections in Asia [5].

Bacteriophages have been proposed to combat *K. pneumoniae* infections. Phage polysaccharide depolymerases represent a promising tool for antimicrobial therapy, and many studies reported the therapeutic effect of the depolymerases. *Klebsiella* giant phage K64-1 was reported to encode nine functional capsule depolymerases for infecting 10 host capsular types, including K1, K11, K21, K25, K30, K35, K64, K69, as well as new capsular types KN4 and KN5 [6]. Solovieva analyzed the polysaccharide-degrading activity of depolymerases from phage KpV71 and phage KpV74, which infect hypervirulent *K. pneumoniae* strains of K1 and K2 capsular types, respectively. The results showed that phage depolymerase Dep_kpv71 was more specific than the bacteriophage itself [7]. These works indicated that phages encoding capsule depolymerases might be potential therapeutic agents; however, the therapeutic effect of such phages against CRKP remains poorly understood. Adsorption is a key stage in bacteriophage recognition of host cells, dependent on the characteristics of receptors on the bacterial surface. As early as the 1970s, *Klebsiella* phage 11 was reported to recognize capsule polysaccharide as the primary receptor for adsorption [8]. Phage NTUH-K2044-K1-1, specific for K1 capsule of *K. pneumoniae*, has demonstrated that capsule is essential for phage infection [9]. Additionally, de Sousa also observed that the capsule plays a major role in shaping *Klebsiella* phage infections, and phages predation stimulates capsule loss [10]. Whereas the interaction between *Klebsiella* phages and capsule serotype has been explored in several studies, the direct interaction between phages and the K47 capsule of *K. pneumoniae* remains unknown.

In the present study, we sought to isolate phages targeting the clinical K47 serotype carbapenem-resistant *K. pneumoniae* (CRKP), investigate key receptors mediating phage-host interactions, and evaluate the potential therapeutic effect. 

## 2. Materials and Methods

### 2.1. Bacterial Strains and Growth Conditions

All *K. pneumoniae* strains were cultured in LB media. When appropriate, antibiotics were used as follows: ampicillin, 100 μg/mL; streptomycin, 100 μg/mL; kanamycin, 50 μg/mL; gentamycin, 20 μg/mL. The differentiation of mucoid and nonmucoid CRKP isolates were identified in BHIA (Difco) with 0.08% (*w*/*v*) Congo red (Sigma-Aldrich, Darmstadt, Germany) supplemented with 5% (*w*/*v*) sucrose and incubated at 37 °C under aerobic conditions for 18 h [11].

### 2.2. Bacteriophage Isolation and Purification

Clinical isolate CRKP A1806 was used for bacteriophage isolation and purification. A modified method of Clokie [12] was applied for the isolation of phages from sewage near hospitals. Phage was harvested from the agar plate and purified three times by picking a single phage plaque. The purified phage was resuspended with SM buffer and stored at 4 °C.

### 2.3. Phage Characterization Assay

#### 2.3.1. Host Range Determination

A total of 47 *K. pneumoniae* strains were spotted directly onto lawns [12]. Briefly, 5 µL of phage stocks (10^9^ PFU/mL) were spotted directly onto lawns of different bacterial strains on LB agar plates, incubated at 37 °C overnight. Test results with positive droplets were confirmed by the formation of plaques of lysis using appropriately diluted phage preparations to obtain efficiency of plating (EOP). 

#### 2.3.2. One-Step Growth Curve and Stability Test

A one-step growth curve was performed to determine the burst size and latent period of the phage [12]. Briefly, mid-log CRKP cells were incubated with the phage at the indicated multiplicities of infection (MOI). After 10 min of phage adsorption, the mixture was centrifuged and suspended in a fresh, cold LB medium. Bacteria cells were incubated aerobically at 37 °C, and samples were taken at 10 min intervals for PFU counts. Burst size was calculated by dividing the average titer of free phages at late time points by the number of initially infected cells. For the thermal stability of phage, aliquots of phage suspensions with high titers (10^8^ PFU/mL) were incubated at 4, 37, 55, and 65 °C for 4 h. For the optimal pH test, phage suspensions were treated with pH 2–12 saline buffers (adjusted with 1 M HCl and 1 M NaOH) at 37 °C for 4 h, respectively. The phage titers were tested by plating on the double-layer agar.

#### 2.3.3. Transmission Electron Microscopy

Propagation of phages was performed using the standard double-layered agar method. Two-microliter droplets of phage suspension (>10^9^ PFU/mL) were placed on copper grids. Negative staining was performed with a uranyl acetate water solution (2% *w*/*v*), and the grids were examined under a transmission electron microscope (TEM) (Hitachi, Tokyo, Japan) at an accelerated voltage of 80 kV. 

### 2.4. Bacteriophage Adsorption and Lytic Activity Assays

Bacteriophage adsorption experiments were conducted at 37 °C in LB medium with slight modification as described previously [13,14]. Briefly, stationary phase CRKP A1806 cells were resuspended in LB medium at the desired concentration. Phage particles (10^7^ PFU/mL) were added to coincubate with host cells (10^8^ CFU/mL) at 37 °C. Samples were withdrawn every 2.5 min and treated with 0.22 μM filters immediately. Unadsorbed phages in the filtrates were determined by titration on the double-layer agar. In order to test phage lysis activity on host cells, phage particles were added into the LB medium containing log-phase CRKP A1806 cells (about 10^8^ CFU/mL) at different MOIs. Samples were withdrawn each 20 min, and OD_600_ was measured.

### 2.5. Phage Genomic Sequencing and Bioinformatics Analysis

Genomic DNA of ΦSRD2021 was purified from phage lysates (10^10^ PFU/mL) using proteinase K and SDS [15] and quantified by Qubit^®^ 2.0 Fluorometer (Thermo Scientific, Waltham, MA, USA). Sequencing libraries were generated using NEBNext^®^ Ultra™ DNA Library Prep Kit for Illumina (NEB, Ipswich, MA, USA) following the manufacturer’s recommendations. The whole genome of ΦSRD2021 was sequenced with Illumina NovaSeq PE150 and assembled with SOAPdenovo software. Assembly results integrated with CISA software were checked with the gap closure software.

Coding sequences were identified and annotated using RAST, BLAST, and Artemis [16,17]. Linear genome comparisons of the phages and visualization of the coding regions were performed with Easyfig 2.2.3 [18]. Phage lifestyle was predicted using PHACTS program [19]. Phage virulence factor analysis was conducted using Virulence Searcher [20]. In order to identify the putative polysaccharide depolymerase, tail fiber protein or hypothetical protein were analyzed by SWISS-MODEL, HMMER, or BlastP. The homologous domain and a typical β-helical structure should be predicted by Phyre2 [21]. For phylogenetic analysis, homologous proteins were identified using BLAST algorithms against the NCBI databases. Multiple alignments were conducted with the ClustalW algorithm, and phylogenetic trees were constructed by MEGA7 software using the neighbor-joining method [22].

### 2.6. Phage Resistant Mutants Screening

To construct the transposon libraries, *E. coli* BW20676 (pRL27) [23] and CRKP A1806 were used as donor and recipient strains, respectively. Kanamycin-resistant transconjugants were grown in LB broth containing a phage of 10^5^ PFU/mL at 37 °C for 4 h. The culture was spread directly onto selected LB agar plates to select survival suspected colonies. In order to confirm the phage resistance of these colonies, suspected candidates were purified by manual streaking onto the plate and examined by spot tests with phage suspensions. The transposon insertion site of the mutants was counted by arbitrary PCR and sequencing [24]. To test the mutation frequency, approximately 10^8^ CFU/mL bacteria cells were mixed with phage particles 10^8^ PFU/mL for 10 min, and the phage-resistant mutants were counted on double-layer agar after 24-h incubation at 37 °C. 

### 2.7. Construction of CPS Mutant

The *wza* insertion mutant was constructed by cloning internal fragments of the *wza* gene into pVIK112 [25]. The resulting plasmids were then introduced into CRKP A1806 by conjugation and integrated into the *wza* locus. The complementation of the *wza* mutant was constructed by cloning fragments of the *wza* gene into pSRKGm. The resulting recombinant plasmids were then introduced into Δ*wza* by conjugation.

### 2.8. Biofilm Formation and Crystal Violet Assay

Overnight cultures of CRKP A1806 were inoculated at 1% into 0.8 mL LB medium containing 0.1% porcine mucin (Sigma-Aldrich, St. Louis, MO, USA) and grown at 37 °C, 80 rpm for 36 h. Formed biofilms in glass tubes were washed four times with sterile water to remove remaining planktonic cells. Into the above tubes, 1 mL fresh LB with or without 10^4^ PFU of ΦSRD2021 was added, and the biofilms were incubated at 37 °C for 4 h of lysing. Biofilms were then rinsed and stained with 0.1% crystal violet and imaged. In order to quantify the biofilms, crystal violet was dissolved with 1 mL DMSO, and OD_570_ was measured [26].

### 2.9. Galleria Mellonella Model

In order to test phage activity for persistence in the hemolymph of uninfected *G. mellonella*, after surface disinfection using 70% ethanol, larvae were injected with 10 μL of phage suspensions in SM buffer (10^8^ PFU/mL) into the last right proleg using a Hamilton syringe with a 50-gauge needle [27]. Equal 10 µL of hemolymph were collected from three worms at 12 h intervals, serially diluted, and plated for quantification. In order to determine a lethal dose of CRKP A1806, a series of 10-fold serial dilutions containing 10^6^ to 10^9^ CFU/mL in PBS were injected into *G. mellonella* larvae. For immediate phage therapy, larvae were injected with 10 µL PBS containing a lethal dose of CRKP A1806 2 × 10^5^ CFU and 2 × 10^3^ PFU ΦSRD2021 simultaneously. For a delayed treatment, larvae infected with 2 × 10^5^ CFU bacteria cells received a single phage injection with 10 µL SM buffer at MOI 0.01 or 0.05, 2 h after CRKP infection [28]. To test the preventive efficacy of pre-phage treatment, the phage (2 × 10^3^ PFU) was administered intramuscularly 6, 12, or 24 h before the challenge with a potentially lethal CRKP A1806 2 × 10^5^ CFU/larvae. Larvae were scored as dead (they did not respond to physical stimuli) or alive 48 h post-infection at 37 °C in the dark. 

### 2.10. The Adult Mouse Model 

Animal experiments were performed in accordance with animal protocols approved by the Ethical Committee of Animal Experiments of Nanjing Agricultural University (permit SYXK [Su] 2017-0007). Five-week-old female CD-1 mice were provided with drinking water containing 0.5% (wt/vol) streptomycin and 0.5% aspartame for 24 h administration. Cultures of CRKP cells were washed and resuspended in PBS overnight, then 100 µL of approximately 10^8^ CFU cells were inoculated into the mice intragastrically. After 12 h of CRKP infection, 100 µL SM buffer containing 10^8^ PFU ΦSRD2021 was intragastrically administered to mice for phage therapy. Fecal pellets were collected from each mouse at the indicated time points, washed thrice with PBS, serially diluted, and then plated on LB plates. Phage titers were determined directly by double-agar-overlay plaque assays. After a 5-day period of incubation, the mice were sacrificed. The small intestines were harvested and homogenized, and the bacteria and phage titer were determined according to the above method.

## 3. Results

### 3.1. Characteristics of Phage SRD2021

To isolate lytic phages for CRKP, we collected sewage water samples near hospitals in different regions of China. A total of 47 CRKP clinical strains were assayed altogether with samples by the spot test (Table 1 and Appendix A). We obtained a phage named ΦSRD2021 that lysed both CRKP A1806 and A1502. A serotyping assay showed that A1806 and A1502 all belong to the K47 serotype. To evaluate the lysis ability of ΦSRD2021, we examined the lytic activity in LB medium at different MOIs. Figure 1A showed that ΦSRD2021 inhibited bacterial growth at all MOIs (OD_600_ of ˂0.1 at 5 h), while the bacterial culture without phage grew rapidly (OD_600_ of 0.4 at 5 h). One-step growth analysis (Figure 1B) showed that ΦSRD2021 had a 10-minute latent period before a burst size of about 80 PFU per infected cell. In the 4 h pH stability tests, the ΦSRD2021 exhibited high stability (>80%) at pH 4 to 10 at 37 °C (Figure 1C). For the optimal thermal tests, a 100% activity at 37 °C, 6.1% at 55 °C, and 0.055% at 65 °C was observed, respectively (Figure 1D). Electron microscopy revealed that ΦSRD2021 had an 80 nm diameter capsid connected with a 180 nm long tail (Figure 1E).

### 3.2. Phage Genome Analysis

DNA sequencing showed that ΦSRD2021 genome is a linear double-stranded DNA of 45,212 bp with a 53.8% G + C content (Genbank accession No. MZ208805). BLASTn analysis revealed that ΦSRD2021 showed high sequence identity to *Klebsiella* phage KP34 (GO413938, 81% query coverage with 91.7% identity) and *Klebsiella* phage vB_KpnP_KpV48 (KX237514, 80% query coverage with 90.9% identity) within the *Autographivirinae* subfamily, *Drulisvirus* genus. Phage SRD2021 was predicted to contain 59 open reading frames (ORFs) on one DNA strand. Like other phages of *Drulisvirus* genus, phage SRD2021 encodes 29 conserved proteins for DNA replication, transcription, packaging, and host lysis (Figure 2A). A putative phage-specific promoter in two locations (1647…1678, 2750…2781) and a Rho-independent transcription terminator (26995…27032), which were highly conserved across all *Drulisvirus* phages, were also discovered in the SRD2021 genome. No lysogenic cycle-related genes, such as an integrase or repressor, or toxin-related genes were identified in the ΦSRD2021 genome. Phylogenetic analyses were performed on amino acid residues of the tail tubular protein B (ORF46 of SRD2021) to investigate the evolutionary relationship between phages from the *Drulisvirus*, *Phimunavirus*, and the more distant phages within the *Autographivirinae* subfamily. Figure 2B clustered ΦSRD2021 and other phages in KP34viruses of the *Drulisvirus* genus into a single monophyletic clade. Additionally, phage capsid assembly scaffolding protein (ORF41) and DNA packaging protein phage terminase large subunit (ORF52) also showed more than 94% sequence identity with phages in KP34viruses of the *Drulisvirus* genus. Overall, these results suggested that ΦSRD2021 may be attributed to KP34viruses of the *Drulisvirus* genus.

### 3.3. Phage SRD2021 Encodes Polysaccharide Depolymerase Gene in Its Tailspike Protein

Phage receptor binding proteins (RBPs), such as tail fiber and tailspike proteins that mediate the initial contact with the receptor on the host cell envelope, often exhibit polysaccharide-depolymerase activity. RBPs are highly specific and a majority of known *Klebsiella* phages have only one or two RBPs. Taxonomically related phages are characterized by a synteny of conserved structural genes interrupted by divergent RBP genes. 

From the genus of KP34viruses, two genes interrupting the synteny of highly conserved structural genes and genes required for phage particle maturation are recognized, and three different groups (A, B, and C) can be further categorized based on differences in length of both genes [29]. Phage SRD2021 contains two large RBPs that are consistent with the characteristics of group B. The first tail fiber RBP (ORF50, 530 aa) showed 79.7% identity (48% query coverage) with the tail fiber protein of *Klebsiella* phage KP34. Two domains were identified in the tail fiber protein: A conserved N-terminal phage binding domain (aa 1–141) belonging to the phage T7 tail superfamily (pfam03906) is responsible for linking the fiber to the tail-tube, and an arsenical pump-driving ATPase domain (TIGR04291) involved in energizing pumps via anchoring into membranes from the cyotosolic face [30]. The second RBP, one predicted hypothetical protein (ORF58, 584 aa), might be corresponding to a capsule depolymerase [29]. It contains an enzymatically active domain that exhibits amino acid homology with the tailspike protein of *Escherichia* phage CBA120 (hydrolase, E-value > 7.6 × 10^−20^) and pectate lyase 3 family protein (PF12708, E-value: 3 × 10^−6^). Among *Drulisvirus* phages of high homology, such as Kp2, vB_KpnP_KpV74, and KP34, the tail fiber and tailspike proteins were more variable than other structural proteins, indicating the high adaptation properties of bacteriophages in modulating their host range during the coevolutionary arms race.

### 3.4. Phage SRD2021 Uses K47 Capsular Polysaccharide as a Receptor

Although several studies described the depolymerases exhibiting CPS-degrading activity within the *Drulisvirus* genus, few works examined the role of the host capsule during phage adsorption and infection. To determine the host receptor of ΦSRD2021, we constructed a CRKP A1806 transposon-insertion library to screen the phage-resistant mutant. Among 10^6^ clones, we obtained one mutant which ΦSRD2021 could not produce any plaque on. Further gene analysis showed that the mutation site was in the *wza* gene which encoded an integral outer membrane protein essential for capsule export. Interestingly, we observed that the *wza* insertion mutant strain was a half-transparent colony on LB agar plate, different from the white opaque colony of the wildtype. 

The Congo red agar (CRA) plate test differentiated the bacteria colony phenotype between mucoid and nonmucoid due to different slime and capsule production. We spread different CRKP cells on the CRA plate to visualize their colony morphology. As Figure 3A showed, the wildtype CRKP cells formed white colonies (left), named mucoid phenotype, whereas the insertion mutant Δ*wza* showed gray colonies (right), named nonmucoid (NM) phenotype. Phage adsorption assay showed that more than 90% of ΦSRD2021 particles adsorbed onto the wild-type host bacteria in 10 min, whereas ΦSRD2021 failed to attach to Δ*wza* host cells (Figure 3B). Similarly, phage SRD2021 could not produce any plaques (EOP < 10^−9^) on the Δ*wza* host cells. In liquid culture lytic activity analysis (Figure 3C), Δ*wza* did not show any decrease in OD_600_ value when infected with ΦSRD2021. By determining the phage titer in liquid culture 2 h after infection, we demonstrated that ΦSRD2021 still failed to propagate in Δ*wza* cells (Figure 3D). The spontaneous mutation rate to ΦSRD2021 resistance was approximately 10^−7^, and all observed phage-resistant mutants produced a nonmucoid colony. Overall, we concluded that ΦSRD2021 shows high specific infectivity against mucoid CRKP, and the capsular polysaccharide serves as a receptor.

### 3.5. Phage SRD2021 Lysed Mature Biofilms Efficiently

When mature biofilms were formed in glass tubes after a 36-h incubation, we replaced the culture with fresh LB with or without phages, and assayed biofilm after 4 h. Figure 4 shows that more than 80% of biofilm was dispersed with 10^5^ PFU/mL of phage, whereas the *wza* mutant was unaffected by phage exposure.

### 3.6. Phage SRD2021 Showed Preventive and Therapeutic Effect in G. mellonella Model

A wax moth larvae model was used to evaluate the therapeutic efficacy of ΦSRD2021 against CRKP A1806 in vivo. Figure 5A showed that the number of phages collected from the hemolymph of larvae injected with ΦSRD2021 were almost stable in a 24-h period but decreased by 8.2-fold at 36 h post-infection. Then, the therapeutic effect of immediate and delayed phage therapy and the preventive effect of pre-phage treatment were assessed in terms of death versus survival. Infected larvae coupled with phage treatment immediately showed 60% survival rates, whereas in the group without phage treatment, all larvae died within 24 h post-infection (Figure 5B), indicating that ΦSRD2021 may reduce the virulence of CRKP in vivo.

The 2-hour-delayed phage treatment test was applied to check the phage therapeutic effect further. So, we tested the phage-delayed therapeutic effect afterward. Figure 5C shows that only the high dose of phage suspensions (MOI 0.05) improved the larvae survival rate with 80% at 48 h post-infection, and the MOI 0.01 of phages could only prolong the survival time of larvae for an extra 12 h. However, when larvae were treated with ΦSRD2021 administration at 6, 12, or 24 h before bacteria challenge, more larvae survived in all pre-phage treatment groups significantly. We also observed that the preventive effect was attenuated over time (Figure 5D). The results showed that the rescue time is the key factor of larvae survival rate, whether in pre-treatment or post-therapy.

### 3.7. Phage SRD2021 Reduced Colonized K. pneumoniae in Mouse Intestine 

Since dense colonization of the intestine by CRKP is associated with an increased risk of bacteremia, we tested the effect of phage treatment in the mouse colonization model. Figure 6 shows that when ΦSRD2021 was fed to the mice at 12 h post-infection, colonized CRKP reduced significantly by 19.7-fold at 1 day compared to the phage free group. This decline rate reached a peak at 2 days post-infection, with up to 131.1-fold decrease of colonized bacteria in feces. Phage titers were 10^6^–10^7^ PFU/g in the first 2 days and reduced to 10^3^–10^4^ PFU/g in day 3~5. Those data suggest that we can take advantage of ΦSRD2021 in eliminating the K47 serotype CRKP in the intestine.

## 4. Discussion

*K. pneumoniae* is an important nosocomial pathogen that causes a wide range of community- and healthcare-associated infections, including pneumonia, bacteremia, urinary tract infections, and even life-threatening septic shock. The emergence of hypervirulent and antibiotic-resistant strains highlights the critical need for effective strategies against this pathogen to prevent its spread [1]. Since there is renewed interest in phages as therapeutic tools, we isolated and characterized a novel *K. pneumoniae* bacteriophage SRD2021 belonging to the genus *Drulisvirus*, targeting K47 serotype *K. pneumoniae* specifically.

Transposon screening showed that ΦSRD2021 lost the capacity of adsorption and infectivity to the *wza* mutant derived from its host CRKP A1806. *Klebsiella* strains with a mutation in *wza* have no capsular polysaccharide and cannot synthesize any detectable intracellular polymer [31]. Likewise, mutants with a nonmucoid phenotype that lose capsule layers or other unknown reasons were also resistant to phage infection. These results indicated that bacterial capsule was indispensable for the adsorption and subsequent infection by ΦSRD2021. Capsule is an important virulence factor of *K. pneumoniae*, and the bacteria exhibit a mucoid appearance due to the robust capsule. Mucoid occurs upon constitutive activation of the sigma factor AlgT/U which regulates the synthesis of the polysaccharide alginate and dozens of other secreted factors. The mucoid bacteria isolated from patients with chronic infections promote coexistence with other pathogens in vivo [32,33], while the phage-resistant mutants losing capsule layers usually have minor virulence [34]. These results indicated the potential benefits of ΦSRD2021 in the treatment of hypervirulent CRKP infection. On the other hand, it is important to note that ΦSRD2021 had no lytic effect on Δ*wza* mutant and other nonmucoid phenotype variants, demonstrating the limitation of single phage therapy. The character and location of the host cell receptors recognized by bacteriophages varies greatly depending on the phage and host. They range from peptide sequences to polysaccharide moieties [35]. Interestingly, in our previous work, the Δ*wza* mutant was sensitive to another *Klebsiella* phage, NJS1, which targets nonmucoid phenotype strains specifically [11]. Compared with a single phage, phage cocktail significantly checked the emergence of resistant mutants [36] and are considered the “drugs” for controlling gastrointestinal CRKP. 

Phages recognizing capsular polysaccharides are described as exhibiting enzymatic activities associated with the tail structures of virions that degrade the capsules, thereby permitting the phage access to the surface of the outer membrane for irreversible binding [37]. The spectrum of ligands to which the tail fibers and tailspikes can bind is the primary determinant of the host range. Our analysis showed that tailspike proteins were variable among members of *Drulisvirus* even though these deeply related phages exhibited the same capsular polysaccharide specificity (K1), such as *Klebsiella* phage vB_KpnP_kpV41, vB_KpnP_kpV71, and NTUH-K2044-K1-1 [7,9]. The predicated tailspike protein of ΦSRD2021 containing pectate lyase 3 family protein domain was homologous to the tailspike TSP2 of *Escherichia* phage Cba120, which involved binding to and digesting the *E. coli* O157 O-antigen [38]. These data suggested the diversity of the tailspikes of *Drulisvirus* in degrading specific capsular type strains. Studies on the interaction of phage Ⅱ with isolated Vi-antigen demonstrated that virion adsorption is accompanied by enzymatic cleavage of side acetyl groups, and deacetylated Vi-polysaccharide, which loses the capacity for further phage reversible binding [39,40]. In another study, the depolymerase encoded by the *Klebsiella* phage K11 aims at depolymerizing the main chain of capsular polysaccharide, and the capsule acts only as a receptor for the initial phage attachment, whereas cell wall components are essential for irreversible binding [8]. ΦSRD2021 could not adsorb onto the *wza* mutant, resulting in the subsequent infection failure; thus, binding to the capsule is essential for ΦSRD2021 successful adsorption and infection. We assume that ΦSRD2021 may attach to the cell surface with hydrolyzing capsule when tail fibers anchor into membranes, successfully triggering phage DNA injection [39].

Biofilm cells exhibit increased resistance to antimicrobials and host immune responses comparable to their planktonic counterparts. Moreover, biofilm formed by pathogens has been implicated as a source of persistent infection and contamination in medical settings [41]. Reports of natural lytic phage with phage-borne polysaccharide depolymerases have shown that phage-induced lysis and exopolysaccharide degradation are used in combination with natural systems to reduce bacterial biofilms [42]. ΦSRD2021 can efficiently lyse both planktonic cells and biofilms of host CRKP A1806. It has been demonstrated that *K. pneumoniae* infection of *G. mellonella* can model some of the known features of *K. pneumoniae*-triggered pneumonia [28]. In this study, we showed that immediate therapy on infected larvae with ΦSRD2021 rose the survival rate of larvae significantly. More importantly, both a pre-phage treatment and a phage-delayed treatment protected larvae effectively. *K. pneumoniae* rectal colonization is a risk factor for mortality in patients with diabetic foot infections [43]. A therapeutic effect was also observed in the mouse colonization model. These data indicated that ΦSRD2021 had a potential preventive and therapeutic effect and may provide a viable alternative therapy to antibiotics in the fight against CRKP intestinal colonization.

## 5. Conclusions

We identified that bacteriophage SRD2021 within *Drulisvirus* genus specifically targets the K47 capsular polysaccharide, supporting the role of capsule for adsorption and infection by bacteriophage encoding capsule depolymerase. The high lysis efficiency on biofilm together with the anti-virulence properties showed in both *G. mellonella* and the mouse colonization model establish the bacteriophage SRD2021 role as a potential therapeutic agent against CRKP infections.

## Figures and Tables

**Figure 1 antibiotics-10-00894-f001:**
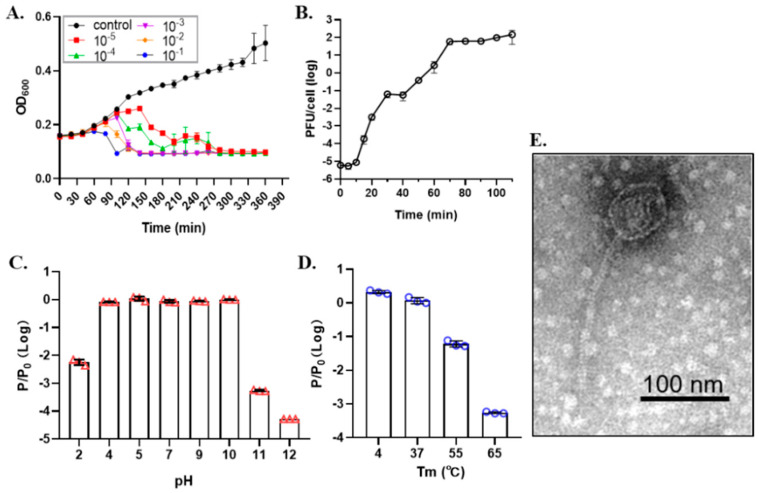
Characteristics of ΦSRD2021. (**A**) ΦSRD2021 infectivity on CRKP A1806 at different MOIs. (**B**) One-step growth curve of ΦSRD2021. (**C**) Thermal stability. Phage suspensions were incubated at 4, 37, 55, and 65 °C for 4 h; (**D**) pH stability. Phage suspensions were treated with different pH buffers at 37 °C for 4 h, respectively. (**E**) Phage morphology of ΦSRD2021 visualized by TEM. The data represent the mean ± standard deviation from triplicate experiments.

**Figure 2 antibiotics-10-00894-f002:**
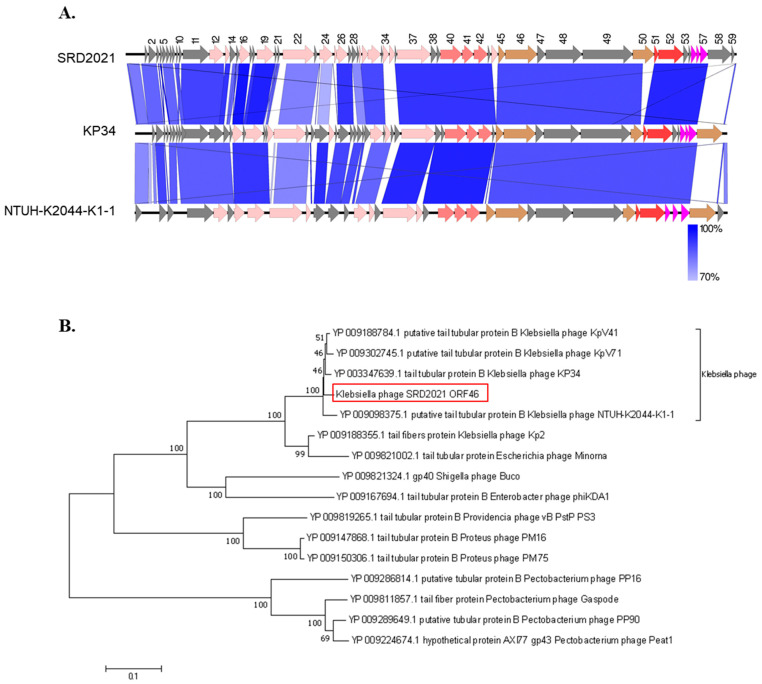
(**A**) Direct comparison of the genomic structures of SRD2021 and close-related members of *Drulisvirus* genus. The color of each gene refers to the functional category: DNA replication and transcription (pale pink), structural protein (capsid: light red; tail: brown), DNA packaging (mid red), and lysis cassette (magenta). (**B**) Phylogenetic tree based on the amino acid residues of the tail tubular protein B by the Neighbor-Joining method.

**Figure 3 antibiotics-10-00894-f003:**
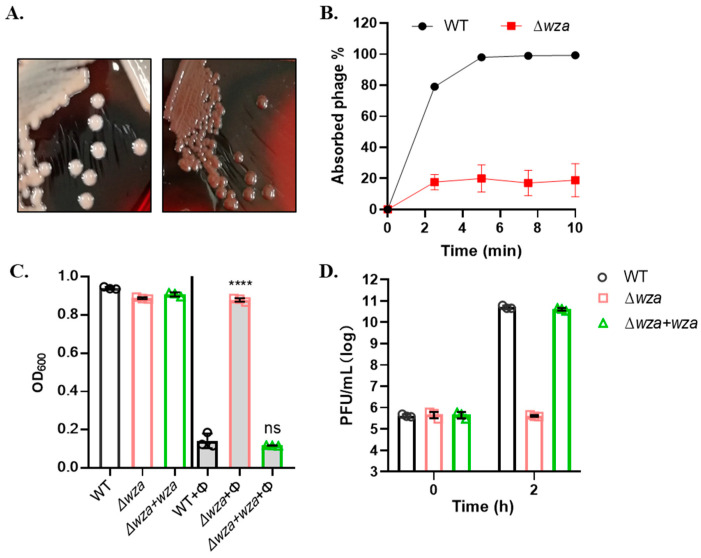
Phage SRD2021 infectivity on CRKP A1806 wildtype and CPS mutant strains. (**A**) Colony of wildtype (**left**) and Δ*wza* (**right**) of CRKP on CRA plate. (**B**) Phage SRD2021 adsorption to wildtype and Δ*wza* mutant, shown as residual PFU percentages. (**C**) The infectivity difference of ΦSRD2021 on wildtype, Δ*wza*, and Δ*wza* + *wza* mutant strains. The host bacteria were infected with ΦSRD2021 at MOI of 0.01. (**D**) Phage titers during ΦSRD2021 infection on wildtype, Δ*wza*, and Δ*wza* + *wza*. All data are the mean of three independent experiments. **** *p* < 0.0001 (2-way analysis of variance [ANOVA]); ns, no significance.

**Figure 4 antibiotics-10-00894-f004:**
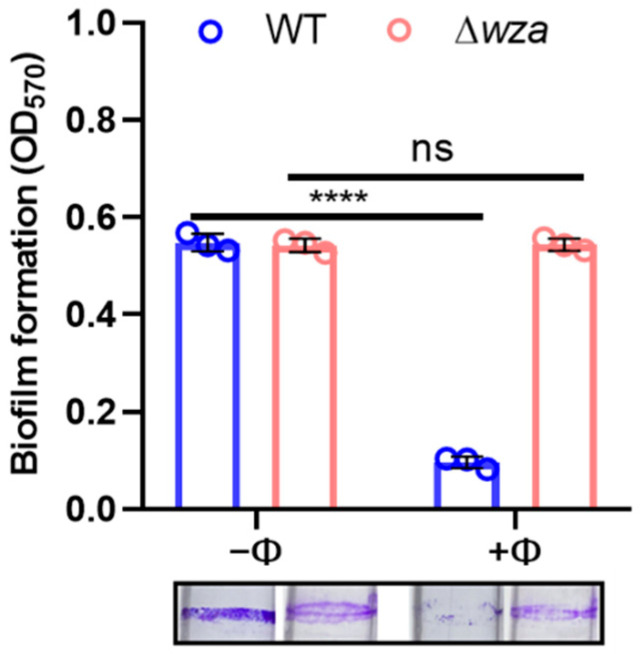
Phage SRD2021 lysis efficiency on formed biofilms of wildtype and Δ*wza* mutant of CRKP A1806. The data represent the mean ± standard deviation from triplicate experiments. **** *p* < 0.0001 (2-way ANOVA); ns, no significance.

**Figure 5 antibiotics-10-00894-f005:**
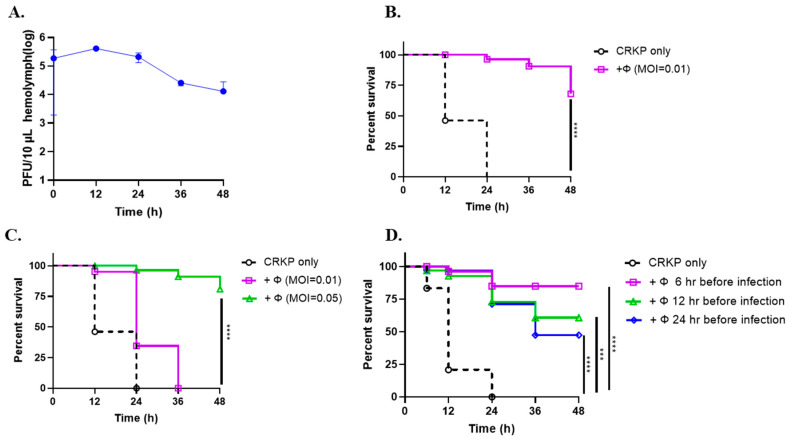
Phage SRD2021 therapy in *G. mellonella* model. (**A**) Bacteriophage persistence in *G. mellonella* larvae. Larvae were injected with 10 µL of phage (10^8^ PFU/mL) in SM buffer. Equal 10 µL of hemolymph was collected from worms at 12-h intervals, and samples were serially diluted and plated for quantification. (**B**) The therapeutic effect of immediate phage treatment. Larvae were injected with 10 µL PBS containing a lethal dose of approximately CRKP A1806 2 × 10^5^ CFU and 2 × 10^3^ PFU ΦSRD2021 simultaneously (*n* = 10 larvae per group). (**C**) Effect of phage-delay treatment on rescuing infected larvae. Larvae infected with approximately 2 × 10^5^ CFU bacteria cells received a single dose of phage injection at MOI 0.01 or 0.05, 2 h after CRKP infection. (**D**) The preventive effect of phage treatment. Larvae were treated with ΦSRD2021 administration (2 × 10^3^ PFU) at 6, 12, or 24 h before bacteria challenge with a lethal dose of CRKP A1806 2 × 10^5^ CFU. Larvae were incubated at 37 °C in the dark and the survival rate was scored every 12 h. *** *p* < 0.0002 (Gehan-Breslow-Wilcoxon test); **** *p* < 0.0001.

**Figure 6 antibiotics-10-00894-f006:**
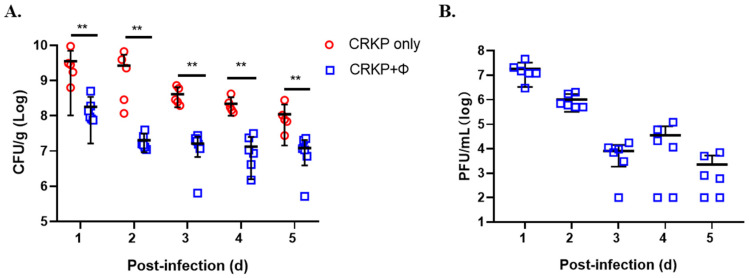
The effect of phage SRD2021 treatment in mouse intestine colonization. (**A**) CRKP CFU. (**B**) Phage titers. Approximately 10^8^ CFU CRKP cells were intragastrically administered. SM buffer or SM buffer containing 10^8^ PFU of phages was administered at 12 h post-infection. Fecal pellets were collected each day and CRKP numbers and phage titers were determined. Horizontal lines represent the average CFU level of CRKP colonized from six mice. ** *p* < 0.01 (Mann-Whitney test).

**Table 1 antibiotics-10-00894-t001:** Characteristics of the clinical K47 serotype CRKP A1806 and A1502.

Strain	MLST ^a^	Colony Phenotype	Characteristics of Bla ^b^	Source
A1806	ST1493	mucoid	NDM-1	unknown
A1502	ST11	mucoid	KPC-2	lower respiratory secretions

^a^ MLST, multilocus sequence typing, ^b^ Characteristics of bla: NDM, New Delhi metallo-β-lactamase; KPC, carbapenemases.

## Data Availability

Data available in a publicly accessible repository.

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
