# Peer review of "Bacteriophage SRD2021 Recognizing Capsular Polysaccharide Shows Therapeutic Potential in Serotype K47 Klebsiella pneumoniae Infections"

_antibiotics, 2021, doi:10.3390/antibiotics10080894_

Round 1
Reviewer 1 Report
This paper sounds scientifically accurate and looks like well written and concise. Due to the importance of K. pneumoniae dependent nosocomial infections - especially in this pandemic period - it is very important to study this topic especially in relation to the types of bacteria resistant to antibiotics.
Furthermore, the result of this study is very promising, since in addition to providing a new line of contrasting CRKP infections, it once again demonstrates the potential of phages in their applications.
I recommend acceptance in the present form and I have just one suggestion: it would be preferable to insert in the body of the text the table that is currently in the supplementaries.
Author Response
Reviewer 1:
This paper sounds scientifically accurate and looks like well written and concise. Due to the importance of K. pneumoniae dependent nosocomial infections - especially in this pandemic period - it is very important to study this topic especially in relation to the types of bacteria resistant to antibiotics.
Furthermore, the result of this study is very promising, since in addition to providing a new line of contrasting CRKP infections, it once again demonstrates the potential of phages in their applications.
I recommend acceptance in the present form and I have just one suggestion: it would be preferable to insert in the body of the text the table that is currently in the supplementaries.
Please NOTE: Due to the conflict of the same registration of phage name, the
GenBank suggested us to rename it as SRD2021 On 17th June. We have replaced it accordingly in the manuscript and rebuttal.
We are very grateful that this reviewer provided kind and encouraging comments. We agree that it is better to demonstrate the characters of strains in the main text. However, since most of strains are negative control that could not be lysed by phage SRD2021, we request to insert the characters of Klebsiella pneumoniae A1806 and A1502, the target lysed strains of phage SRD2021, in the main text as Table 1, and present the rest of nonlysed strains information in the Supplemental Materials.
Reviewer 2 Report
The manuscript by Hao and colleague provides a great characterization of a bacteriophage infecting Klebsiella pneumoniae and its therapeutic potential. I found it to be particularly relevant to characterize the activity of the phage in in vitro (planktonic, biofilm) and in vivo (G. melonella, mice) activity, both in preventive and therapeutic approaches. I only have minor comments and suggestions as detailed below.
Minor comments
- Considering the focus on the therapeutic potential of the phage, I suggest that the authors indicate somewhere in the text if the phage has or has not any gene suggestive of a lysogenic life cycle.
- Line 106, what is the composition of the buffer(s) used to test the pH? Did you use different buffers or the same buffer at different pH values?
- Line 116, why were stationary cells used for this assay?
- Lines 119 and 123, which time points?
- Lines 149-150, it is unclear how you distinguished between phage susceptible and phage resistant mutants. It would be useful to have some more detail in the protocol.
- Figure 1B, the y axis should represent the values in pfu/cell. From this graph, the latent period seems to be 10 min and not 5. How was the burst size calculated?
- Figure 1C and 1D, please plot also the individual values, as you show in Figure 3C.
- Figure 1E, is it possible to make it bigger? It is quite hard to see the phage features as it is.
- Figure 2A, if possible, use a colour-blind-friendly colour palette.
- Figure 2B, do the other proteins also cluster according to phage genus? Please add this information to the figure.
- Line 258, could you give a more clear explanation to why phage φZX1 belongs to group B KP34viruses?
- Figure 3B, it would be preferable to show the % of adsorbed phage instead.
- Figure 3D and Figure 4, show individual points as in Figure 3C.
- Lines 396-397, could the authors discuss about why the phage is not able to bind to the secondary receptor in the cell surface in the wza mutant strain, since in the mutant this receptor is no longer hidden by the capsule and should therefore be more accessible to the phage.
- Line 416, I suggest avoiding stating that it provides new evidence on the role of capsule for adsorption and infection, since this has been shown multiple times in previous studies.
Minor corrections
- Line 90, correct to “The purified phage was resuspended with SM buffer and stored at 4°C”.
- Line 94, correct to “were spotted directly onto lawns”.
- Line 149, correct to “mutants were counted”.
- Line 175, correct to “cells received a single phage injection”.
- Line 198, correct to “A total of 47 CRKP clinical strains”.
- Line 201, correct to “the lysis ability of φZX1”.
- Line 229, correct to “were also discovered”.
- Line 253 misses a reference.
- Throughout the text, make sure to use Drulisvirus with italics and capitalized first letter.
- Line 372, correct to “that lose capsule layers”.
- Line 376, correct to “suggesting that phage cocktail”.
- Line 392, correct to “aim at depolymerizing the main chain”. Also clarify the main chain of what.
- Line 411, correct to “effect and it may provide a viable alternative”.
- Line 415-416, correct to “specifically targets the K47 capsular polysaccharide”.
Author Response
Reviewer 2:
The manuscript by Hao and colleague provides a great characterization of a
bacteriophage infecting Klebsiella pneumoniae and its therapeutic potential. I found it
to be particularly relevant to characterize the activity of the phage in in
vitro (planktonic, biofilm) and in vivo (G. melonella, mice) activity, both in preventive
and therapeutic approaches. I only have minor comments and suggestions as
detailed below.
Minor comments
Considering the focus on the therapeutic potential of the phage, I suggest that
the authors indicate somewhere in the text if the phage has or has not any
gene suggestive of a lysogenic life cycle.
Please NOTE: Due to the conflict of the same registration of phage name, the
GenBank suggested us to rename it as SRD2021 On 17th June. We have
replaced it accordingly in the manuscript and rebuttal.
We are very grateful that this reviewer providing insightful comments to help
make the manuscript better.
Analysis of phage lifestyle and virulence factor showed that lysogenic cycle
and toxin genes were not included in ΦSRD2021 genome. We have added the
predicted results in the manuscript (Line 141-142 and 246-247).
Line 106, what is the composition of the buffer(s) used to test the pH? Did
you use different buffers or the same buffer at different pH values?
Sorry for the unclear description. We used the same solution to test the pH
stability of phage, that is 0.9% saline adjusted with 1M HCl and 1M NaOH to
different pH values. We have added the information.
Line 116, why were stationary cells used for this assay?
We thank this reviewer to raise this question. We conducted phage
adsorption assay based on the references which used stationary phase cells
(ref. 13 and 14). We also assayed it with both mid-log phase and stationary
phase cells, and found that they have similar dynamics which support the
conclusion “more than 90% of ΦZX1SRD2021 particles adsorbed onto the
wildtype host bacteria in 10 minutes”.
Lines 119 and 123, which time points?
We have added the exact time to make the protocol clearer. Please see line 125
and 129.
Lines 149-150, it is unclear how you distinguished between phage susceptible
and phage resistant mutants. It would be useful to have some more detail in
the protocol.
We are sorry for the confusion. More details were added in the protocol (line
152 to 158).
Figure 1B, the y axis should represent the values in pfu/cell. From this graph,
the latent period seems to be 10 min and not 5. How was the burst size
calculated?
We are sorry for the time mistake. Burst size were calculated by dividing the
average titer of free phages at late timepoints by the number of initially
infected cells. We have modified the graph (Fig.1B) to present the results of
both 5 and 10 min, and confirmed that the latent period of phage SRD2021
should be 10 min.
Figure 1C and 1D, please plot also the individual values, as you show in
Figure 3C.
Added as suggested.
Figure 1E, is it possible to make it bigger? It is quite hard to see the phage
features as it is.
Thanks for the suggestion. We have re-arranged Figure 1 to zoom in on 1E as
suggested.
Figure 2A, if possible, use a colour-blind-friendly colour palette.
The reviewer's advice was very thoughtful. We have replaced the potentially
controversial yellow and green color with blue and purple in Figure 2A.
Figure 2B, do the other proteins also cluster according to phage genus? Please
add this information to the figure.
We explored it and found that Phage capsid assembly scaffolding protein
(ORF41) and DNA packaging protein phage terminase large subunit (ORF52)
also showed more than 94% sequence identity with that of phages in
KP34viruses, Drulisvirus genus. The analysis has been added in the main text
(line 252-255).
Line 258, could you give a more clear explanation to why phage φZX1
belongs to group B KP34viruses?
We are sorry for the confusion. Briefly, sequencing analysis indicated that
phage SRD2021 belons to KP34viruses in paragraph 2.2. We further classified
the phage as group B based on the size of two conserved RBPs. In the
revision, we have added the characters of KP34viruses RBPs, and rearranged
the order of the statements to make it easier to understand (Line 266-276).
Figure 3B, it would be preferable to show the % of adsorbed phage instead.
We agree with it. Changed accordingly.
Figure 3D and Figure 4, show individual points as in Figure 3C.
Added as suggested.
Lines 396-397, could the authors discuss about why the phage is not able to
bind to the secondary receptor in the cell surface in the wza mutant strain,
since in the mutant this receptor is no longer hidden by the capsule and
should therefore be more accessible to the phage.
We thank this reviewer for the discussion. The knockout of wza gene on
CRKP A1806 leaded to the failure adsorption of Phage SRD2021 (Fig. 3B), and
the subsequent failed infection. That is, binding to the capsule is essential for
ΦSRD2021 successful adsorption and infection. We assume that ΦSRD2021
may attach to the cell surface with hydrolyzing capsule until tail fibers anchor
into membranes to successfully trigger phage DNA injection (Ref. 39). We
also inserted the hypothesis in the discussion (Line 439-443).
Line 416, I suggest avoiding stating that it provides new evidence on the role
of capsule for adsorption and infection, since this has been shown multiple
times in previous studies.
We thank this reviewer for the rigorous approach. We have modified it to
avoid the misleading.
Minor corrections
Line 90, correct to “The purified phage was resuspended with SM buffer and
stored at 4°C”.
Line 94, correct to “were spotted directly onto lawns”.
Line 149, correct to “mutants were counted”.
Line 175, correct to “cells received a single phage injection”.
Line 198, correct to “A total of 47 CRKP clinical strains”.
Line 201, correct to “the lysis ability of φZX1”.
Line 229, correct to “were also discovered”.
The above have been corrected.
Line 253 misses a reference.
Ref. 30 has been added (Line 283).
Throughout the text, make sure to use Drulisvirus with italics and capitalized
first letter.
Line 372, correct to “that lose capsule layers”.
Line 376, correct to “suggesting that phage cocktail”.
The above have been corrected.
Line 392, correct to “aim at depolymerizing the main chain”. Also clarify the
main chain of what.
We have corrected it to “aim at depolymerizing the main chain of capsular
polysaccharide”.
Line 411, correct to “effect and it may provide a viable alternative”.
Line 415-416, correct to “specifically targets the K47 capsular polysaccharide”.
The above have been corrected.
Reviewer 3 Report
Hao G. et al have identified and characterized the therapeutic potential of bacteriophage ZX1 as a treatment strategy for CRKP. Their work highlights the potential of using bacteriophages as a treatment strategy, and bacteriophage ZX1 could be used as a therapeutic approach for treating K47 serotype CRKP. I have few comments regarding the study as follows:
Animal experiments: In colonization model studies, it has been shown that CRKP disseminates to different GI organs like the liver and spleen. What is the effect of bacteriophage treatment on the dissemination rate in treated versus control groups? CRKP transmissions from host to host happens from its intestinal reservoir (31907407, 32839189) therefore, authors should fix tissue samples and gram stain the intestinal tissue sections to observe the change in bacterial colonies in the small and large intestine post-treatment.
Statistics on data: Some of the graphs(Fig 3B, Fig 5, and Fig 6A) don't have SD or SE showing with the data points.
Capsule shedding has been observed in K. pneumoniae (PMID: 31473336) and S. pneumoniae (PMID: 28830943). Capsule shedding is associated with increased resistance and has no impact on the colonization pathogenicity. Since bacteriophage ZX1 has no effect on Δwza strain, and the mutant strain switched from mucoid to nonmucoid phenotype. In the discussion, the authors should highlight the possible failures of bacteriophage therapy with respect to capsule shedding.
Minor comments
- Line 69- replace was with has
- Line 259- capitalize D of drulisvirus
- Improve the clarity or if there is a color image available for Fig 4 biofilm, please use that.
- SD is not visible in Fig 6B, kindly choose a different color
- Line 371- in vivo is in red, change it to black
Author Response
Reviewer 3:
Hao G. et al have identified and characterized the therapeutic potential of
bacteriophage ZX1 as a treatment strategy for CRKP. Their work highlights the
potential of using bacteriophages as a treatment strategy, and bacteriophage ZX1
could be used as a therapeutic approach for treating K47 serotype CRKP. I have few
comments regarding the study as follows:
Animal experiments: In colonization model studies, it has been shown that CRKP
disseminates to different GI organs like the liver and spleen. What is the effect of
bacteriophage treatment on the dissemination rate in treated versus control groups?
CRKP transmissions from host to host happens from its intestinal reservoir
(31907407, 32839189) therefore, authors should fix tissue samples and gram stain the
intestinal tissue sections to observe the change in bacterial colonies in the small and
large intestine post-treatment.
Please NOTE: Due to the conflict of the same registration of phage name, the
GenBank suggested us to rename it as SRD2021 On 17th June. We have replaced it
accordingly in the manuscript and rebuttal.
We thank this reviewer very much for providing insightful comments to help make
the manuscript better.
It is very meaningful to test the effect of phage therapy on K. pneumoniae dissemination.
We examined the dissemination ability of CRKP A1806. However, we found that this
strain was hard to disseminate to the liver and spleen in our colonization model,
different from K. pneumoniae isolates used in references (31907407, 32839189). The
result may due to the characteristics of strains, routes of infection and host immune
system. Some K. pneumoniae strains could not translocate from the gastrointestinal tract
to extraintestinal sites unless in appropriate animal models, for example exhibiting
damage to the mucosal epithelium (doi.org/10.1007/978-1-4615-4143-1_2). Besides, as
suggested, we observed the phenotype of colonies from intestine post-treatment on
BHIA medium with Congo red (Materials and methods 2.1). Nonmucoid mutant could
be identified, which was consistent with in vitro results, but the transformation
frequency was vary greatly among different mouse. In addition, our phage treatment
was not continuous, thus the conclusion may not be enough to present in this work.
We hope further studies focusing on phage therapy effect on CRKP dissemination in
the future could explore this question deeply and benefit the evaluation of phage
treatment.
Statistics on data: Some of the graphs(Fig 3B, Fig 5, and Fig 6A) don't have SD or SE
showing with the data points.
We sorry for the inaccuracy. We have modified them.
Capsule shedding has been observed in K. pneumoniae (PMID: 31473336) and S.
pneumoniae (PMID: 28830943). Capsule shedding is associated with increased
resistance and has no impact on the colonization pathogenicity. Since bacteriophage
ZX1 has no effect on Δwza strain, and the mutant strain switched from mucoid to
nonmucoid phenotype. In the discussion, the authors should highlight the possible
failures of bacteriophage therapy with respect to capsule shedding.
We thank this reviewer for the suggestion. We have discussed in the previous
version the necessity of phage cocktail application, but not highlighted the short of
single phage therapy. In the revision, we have emphasized the possible failures of
bacteriophage therapy (Line 408-417).
Minor comments
1. Line 69- replace was with has
2. Line 259- capitalize D of drulisvirus
3. Improve the clarity or if there is a color image available for Fig 4 biofilm,
please use that.
4. SD is not visible in Fig 6B, kindly choose a different color
5. Line 371- in vivo is in red, change it to black
Thanks a lot for the details. The above have been corrected.
Round 2
Reviewer 3 Report
The authors have successfully incorporated all suggestions in the manuscript and have answered all queries adequately.